# A genome-wide association study identifies a locus associated with knee extension strength in older Japanese individuals
Shuji Ito[1,2,3], Hiroshi Takuwa[1,3], Saori Kakehi[4,5], Yuki Someya[5,6], Hideyoshi Kaga[4], Nobuyuki Kumahashi[7], Suguru Kuwata[3], Takuya Wakatsuki[3], Masaru Kadowaki[3], Soichiro Yamamoto[3], Takafumi Abe[8], Miwako Takeda[8], Yuki Ishikawa[2], Xiaoxi Liu[2], Nao Otomo[1,2,9], Hiroyuki Suetsugu[1,2,10], Yoshinao Koike[1,2,11], Keiko Hikino[12], Kohei Tomizuka[2], Yukihide Momozawa[13], Kouichi Ozaki[14], Minoru Isomura[8,15], Toru Nabika[8,16], Haruka Kaneko[17], Muneaki Ishijima[5,17], Ryuzo Kawamori[4,5], Hirotaka Watada[4,5], Yoshifumi Tamura[4,5], Yuji Uchio[3], Shiro Ikegawa[1,2] & Chikashi Terao[2,18,19] ✉

Sarcopenia is a common skeletal muscle disease in older people. Lower limb muscle strength is a good predictive value for sarcopenia; however, little is known about its genetic components. Here, we conducted a genome-wide association study (GWAS) for knee extension strength in a total of 3452 Japanese aged 60 years or older from two independent cohorts. We identified a significant locus, rs10749438 which is an intronic variant in *TACC2* (transforming acidic coiled-coil-containing 2) ($P = 4.2 \times 10^{-8}$). *TACC2*, encoding a cytoskeleton-related protein, is highly expressed in skeletal muscle, and is reported as a target of myotonic dystrophy 1-associated splicing alterations. These suggest that changes in TACC2 expression are associated with variations in muscle strength in older people. The association was consistently observed in young and middle-aged subjects. Our findings would shed light on genetic components of lower limb muscle strength and indicate *TACC2* as a potential therapeutic target for sarcopenia.

Sarcopenia is a common skeletal muscle disease in older people, which leads to unfavorable outcomes such as falls, fractures and death[1–3]. Sarcopenia can be defined by using muscle mass, muscle strength and physical performance[4–6]. The loss of muscle mass has mainly been used as an indicator of sarcopenia; however, recent studies suggest that muscle strength is a better indicator that reflects adverse health outcomes of sarcopenia[2,7–9]. For example, Schaap et al. described that low handgrip strength was associated with incidence of falling, independent of a muscle mass[2]. Therefore, muscle strength, rather than a muscle mass, is adopted as the primary indicator for sarcopenia in the revised European Working Group on Sarcopenia in Older People (EWGSOP) algorithm[5].

As an indicator for muscle strength, handgrip strength is commonly used; however, it only reflects the strength of the upper extremities, not that of lower extremities. A recent study reported that knee extension strength, a proxy of lower limb strength, is more strongly associated with performance-based sarcopenia compared to handgrip strength[9,10]. Correspondingly, Yeung et al. reported stronger association of knee extension strength with health characteristics than handgrip strength[11]. Thus, knee extension strength would be a more appropriate indicator of sarcopenia than handgrip strength.

Muscle strength has been known to be heritable[12–14]. Family studies showed the heritability of handgrip strength was 56%[12] and genome-wide association studies (GWASes) of handgrip strength suggested the heritability was 13–24%[15,16]. Previous GWASes have discovered 170 variants associated with muscle strength[15–19]. One GWAS on maximum handgrip strength divided by weight in UK Biobank participants identified 101 loci and showed a shared genetic etiology of handgrip with cardiometabolic and cognitive health[15]. Another GWAS of muscle weakness based on handgrip strength in Europeans aged 60 years or older identified 15 loci[17]. However, there is only one GWAS of lower limb muscle strength, which did not

identify any significant loci[18] due to a lack of statistical power. GWASes of lower limb muscle strength with adequate sample size would add good information for understanding of the genetic architecture of sarcopenia.

In the present study, we performed a GWAS of knee extension strength using 3452 participants aged 60 years or older from two independent cohorts. We identified a locus with genome-wide significance, which has not been identified in previous GWASes of muscle strength. In the locus, we identified a candidate susceptibility gene, *TACC2* (transforming acidic coiled-coil-containing 2) which is highly expressed in skeletal muscle. We also identified several suggestive loci, which include promising candidate susceptibility genes. The variants associated with handgrip strength in a previous GWAS[15] showed an association in our dataset.

## Results

### Sample source, genotyping and imputation

The study design was illustrated in Supplementary Fig. 1. Three sets of samples from two independent cohorts were enrolled in this study, which consisted of a total of 3478 participants aged 60 years or older (Table 1). Set 1 and Set 2 consisted of 1014 and 841 participants. Both were from Shimane CoHRE Study[20,21], but different arrays were used for genotyping. Set 3 consisted of 1623 participants from Bunkyo Health Study[22]. While the methods to measure knee extension strength were slightly different between the two study cohorts, the patterns of distributions of knee extension strength were very consistent (Supplementary Figs. 2, 3).

We used Illumina HumanOmniExpressExome BeadChip for genotyping 1014 participants in Set 1. After quality controls of samples and variants, data on a total of 1007 samples remained for the further analysis (Supplementary Fig. 1). We conducted a whole-genome imputation using an in-house reference panel containing a total of 3256 Japanese whole-genome sequence data and 2504 individuals in the 1000 Genomes Project (1KG phase 3v5). We set a threshold of Rsq more than 0.3 for variants to be included in the following analyses. An association analysis was conducted by the linear mixed model using fastGWA[23]. The top three principal components (PCs) were used as covariates. Association results in this data set are shown in Supplementary Fig. 4.

We used Illumina Asian Screening Array for genotyping 841 and 1623 participants in Set 2 and 3, respectively. After quality controls, 838 and 1607 samples remained in Set 2 and 3, respectively (Supplementary Fig. 1). We conducted the whole-genome imputation as described above. Results in each data set were shown in Supplementary Figs. 5 and 6.

### Genome-wide association studies and a meta-analysis

Then, we conducted a meta-analysis of three GWASes using the fixed-effect inverse-variance weighted method with the use of the METAL software[24]. We took an intersection of imputed variants across the three data sets, resulting in 9,146,474 autosomal variants and 197,639 chromosome X variants. The Manhattan plot and Q-Q plot of the meta-analysis are shown in Fig. 1. We did not find an inflation of statistics (inflation factor ($\lambda_{GC}$) of 1.02) and linkage disequilibrium score regression (LDSC) revealed an intercept of 1.00 (SE, 0.0071), indicating that the current results were not confounded by

## Table 1 | Characteristics of the subjects aged 60 years or older

| | Set 1 | Set 2 | Set 3 |
|---|---|---|---|
| Source of samples | Shimane CoHRE Study (1st cohort) | Shimane CoHRE Study (2nd cohort) | Bunkyo Health Study |
| Number of samples | 1007 | 838 | 1607 |
| Sex (male/female) | 312/695 | 296/542 | 678/929 |
| Mean age (s.d.) | 72.8 (6.7) | 72.0 (6.4) | 73.1 (5.4) |
| Mode of knee extension strength | isometric | isometric | isokinetic |
| Device | QTM-05F | QTM-05F | BIODEX |
| Mean muscle strength/ Body weight (s.d.) | 0.644 (0.212) kg/kg | 0.563 (0.186) kg/kg | 1.332 (0.375) Nm/kg |
| Genotyping platform | Ilumina, OmniExpressExome | Ilumina, Asian Screening Array | Ilumina, Asian Screening Array |

"Number of samples" is after quality control. s.d., standard deviation.

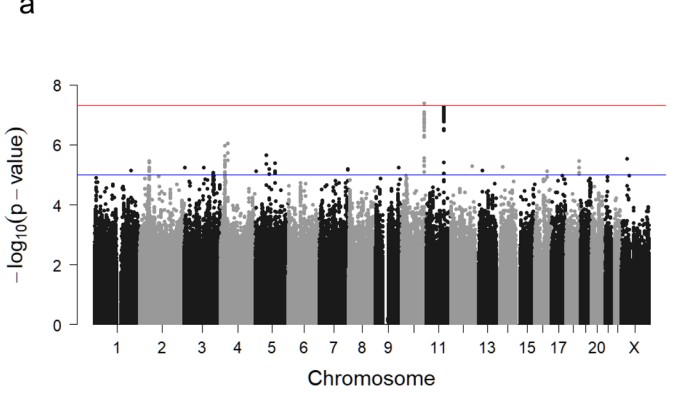
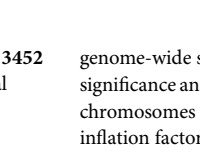
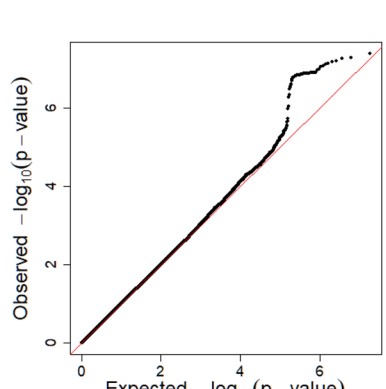

**Fig. 1 | A genome-wide association analysis of knee extension strength in 3452 participants aged 60 years or older. a** Manhattan plot. *X*-axis: chromosomal location. *Y*-axis: $-\log_{10}$ p-value for each genetic variant. Horizontal red line: genome-wide significance ($P < 5 \times 10^{-8}$). Horizontal blue line: suggestive genome-wide significance ($P < 1 \times 10^{-5}$). A locus with genome-wide significance and a locus with suggestive genome-wide significance were identified on chromosomes 10 and 11, respectively. **b** Q-Q plot for the analysis. The genomic inflation factor ($\lambda_{GC}$) was 1.02.

any bias and no apparent strong polygenic effects on the muscle strength in lower limbs presumably due to the limited sample size in the current study. A heritability calculated by LDSC was 8.9%, indicating a substantial contribution of genetic components on the lower limb muscle strength.

### A novel locus associated with knee extension strength

We identified a novel locus significantly associated with knee extension strength (Fig. 1 and Table 2). The lead variant is rs10749438, an intronic variant in *TACC2* (transforming acidic coiled-coil-containing 2) located at 10q26 (Beta = −0.15, $P = 4.2 \times 10^{-8}$) (Fig. 2a and Table 2). The risk allele for muscle weakness is allele A. We observed consistent associations of this variant across the three data sets and no heterogeneity of association results was observed ($I^2 = 0$, Fig. 3). rs10749438 was positioned at enhancer-like histone marks, H3K27ac in skeletal muscle according to the ENCODE database[25] and HaploReg (v4.1)[26]. A statistical fine-mapping analysis revealed rs10749438 with the highest posterior probability (Supplementary Table 1). *TACC2* encodes a cytoskeleton-related protein[27] that concentrates at centrosomes throughout the cell cycle and is reported as a target of myotonic dystrophy 1-associated splicing alterations[28]. Skeletal muscle showed high *TACC2* expression according to Genotype-Tissue Expression project version 8 (GTEx v8)[29] (Supplementary Fig. 7).

Since *TACC2* was reported to be susceptible to transcriptional regulation effect of androgen receptor[30] and our three data sets were female-dominant, we conducted sex-specific analysis and observed the effect size and direction of rs10749438 are consistent regardless of sex (Table 3 and Supplementary Tables 2, 3). The age-stratified analysis also showed that effect directions were consistent (Table 3 and Supplementary Table 4). We further conducted analyses using a total of 173 subjects aged under 60 years to evaluate whether this association was specific to older people or not (Supplementary Fig. 8 and Supplementary Table 5). The analysis of the participants aged under 60 years also showed a consistent trend of the association (Table 3 and Supplementary Table 6). While the effect size tended to be strong in the participants aged 75 years or older in comparison with that aged 60 years or older, the trend was not held in subjects under 60 years. The meta-analysis revealed a further increased association of this variant ($P = 1.2 \times 10^{-8}$, $I^2 = 0$; Table 4 and Supplementary Fig. 9), suggesting that the association might be observed in a general population.

### Suggestive loci associated with knee extension strength

We also identified 17 suggestive loci ($P < 1.0 \times 10^{-5}$, Table 2). Among these loci, rs1718074 in an intron of the dystrophin gene (*DMD*) located at Xp21.2-p21.1 is the most noteworthy (Beta = 0.14, $P = 2.9 \times 10^{-6}$) (Table 2). *DMD* is the disease gene for Duchenne muscular dystrophy and Becker muscular dystrophy, both of which show progressive deterioration of muscle tissue and resultant weakness[31]. The effect size and the effect direction of rs1718074 were consistent between males and females (Supplementary Tables 2 and 3).

rs6483495, an intronic variant of *MAML2* located at 11q21, showed the borderline significant *p*-value (Beta = −0.14, $P = 5.4 \times 10^{-8}$) (Fig. 2b and Table 2). The locus has not been reported in muscle-related GWASes. Skeletal muscle does not highly express *MAML2*. While we observed consistent associations of this locus across the three data sets ($I^2 = 0$, Table 2), when we expanded the participants to those aged under 60 years, we observed the opposite direction compared to the result of older participants (Supplementary Table 6). These findings suggest that further studies are necessary to confirm the association between *MAML2* and knee extension strength.

### Evaluation of the variants identified in a GWAS of handgrip strength

We further tested whether the variants associated with handgrip strength[15–17] showed associations in our dataset. Out of 170 variants associated with three GWASes of handgrip strength, 132 were included in our dataset, and 18 proxy variants were selected for the test based on the linkage disequilibrium (LD) of Europeans. Among the 150 variants, 87 showed the same direction of effect and a binomial test *p*-value was 0.03. Among the 87 variants, five showed an association of nominal statistical significance (expected number: 4.35). These findings suggest that muscle strength of upper and lower limbs may share a small part of genetic architecture.

## Discussion

We conducted the GWAS of knee extension strength. There are a few limitations to our study. First, the sample size is not large enough to detect many genome-wide significant loci. Future studies with large sample sizes would be necessary. Second, we used different devices to measure the knee extension strength in the two cohorts; that is, one used isometric testing and the other isokinetic testing. We believe that the impact of the difference of the testings on the association is expected to be small since these testings are reported to be highly correlated[32]. Their distributions were very consistent (Supplementary Figs. 2, 3).

Between knee extension strength and handgrip strength, a poor to moderate correlation has been reported[9,33–35], which could explain their substantial but relatively weak shared directions of effects. Accordingly, our results suggest that only small fraction of genetic architecture is shared between muscle strength of upper and lower limbs. While lower limb muscle strength is reported to be more strongly associated with sarcopenia than handgrip strength[9–11], most GWASes for muscle strength were based on handgrip strength[15,16]. Thus, our identification of the genetic variants associated with lower limb strength would shed light on the etiology of sarcopenia.

We successfully identified the locus with the genome-wide significance, which contained a candidate gene, *TACC2*. The association was the same as a meta-analysis with the random effect model (Beta = −0.148, $P = 4.2 \times 10^{-8}$). We additionally investigated if there is a possible confounding effect of knee osteoarthritis on knee extension muscle strength by using knee osteoarthritis as an additional covariate, but we found no confounding effect of knee osteoarthritis (Beta = −0.15, $P = 3.7 \times 10^{-8}$). To the best of our knowledge, the present study is the first GWAS for lower limb muscle strength that identified a significant locus. In line with our results, a previous GWAS of muscle weakness based on handgrip strength in European elderly showed the consistent association between rs10749438 and muscle weakness with nominal statistical significance ($P = 0.037$)[17]. We also investigated if rs10749438 is associated with other sarcopenia-related traits such as lean body mass[36], frailty[37], walking pace[38], fatigue[39], testosterone[40] and IGF1[41] in the UK Biobank, which did not show any nominal significant association. rs10749438 is located at enhancer-like histone marks, H3K27ac in skeletal muscle and *TACC2* is highly expressed in skeletal muscle. While the top posterior probability and overlapping with the enhancer region suggest rs10749438 as a promising candidate of a causal variant, functional follow-up is necessary to conclude this point. Regarding a responsible gene in this association, using cell cultures from human embryonic muscle, myotonic dystrophy 1-associated splicing alterations were significantly enriched in *TACC2* which is one of cytoskeleton-related gene[27,28]. Although the variants are not an expression quantitative trait locus (eQTL) for *TACC2* according to GTEx v8[29], there is a possibility that the variant's functional effect is more context-dependent. In fact, sampling site of muscle in GTEx is not quadriceps femoris muscle but gastrocnemius muscle. These findings suggest that *TACC2* is a good candidate gene for muscle strength and further experimental validation using animal models will be needed. Another possibility is that the variant regulates other distant genes, *ATE1*, *NSMCE4A* and *BTBD16*. These genes are also candidate of causal genes for knee extension strength. Additionally, there is a possibility that the variant's functional effect on *TACC2* or another gene is more context-dependent and existing eQTL studies may not have detected such effects yet.

TACC2 belongs to the TACC protein family which involves in the complex process of regulating microtubule dynamics during cell division[42]. TACC genes lie within a chromosomal region associated with tumorigenesis. Mammalian TACC proteins, namely TACC1, TACC2 and TACC3 interact with microtubules, and control cell growth and differentiation during cell division[43]. Several studies indicated high expression of *TACC2*

**Table 2 | Significant and suggestive loci associated with knee extension strength in 3,452 participants aged 60 years or older**

| rsID | Chr | Position | Gene | Location | Ref | Alt | META | | | Set 1 | | Set 2 | | Set 3 | | P_het | I² | INFO |
|---|---|---|---|---|---|---|---|---|---|---|---|---|---|---|---|---|---|---|
| | | | | | | | Freq | Beta | P-value | Beta | P-value | Beta | P-value | Beta | P-value | | | |
| rs141279361 | 1 | 202923428 | ADIPOR1 | 5'UTR | C | T | 0.026 | −0.356 | 7.08E-06 | −0.413 | 4.35E-03 | −0.222 | 1.52E-01 | −0.397 | 9.05E-04 | 0.599 | 0 | 0.91 |
| rs77607073 | 2 | 49531177 | FSHR/NRXN1 | intergenic | A | G | 0.115 | 0.177 | 3.47E-06 | 0.101 | 1.61E-01 | 0.281 | 1.44E-04 | 0.164 | 4.01E-03 | 0.207 | 36.5 | 0.98 |
| rs147964289 | 3 | 3825426 | CRBN/LRRN1 | intergenic | A | C | 0.011 | −0.556 | 5.80E-06 | −0.644 | 2.34E-03 | −0.591 | 1.75E-02 | −0.466 | 1.38E-02 | 0.811 | 0 | 0.85 |
| rs141616911 | 3 | 105393996 | CBLB | intronic | T | C | 0.008 | 0.680 | 5.86E-06 | 0.765 | 2.03E-02 | 0.724 | 2.20E-02 | 0.631 | 1.53E-03 | 0.929 | 0 | 0.85 |
| rs138068168 | 3 | 158936130 | IQCJ/IQCJ-SCHIP1 | intronic | T | TTG | 0.476 | −0.115 | 8.30E-06 | −0.165 | 4.00E-04 | −0.053 | 3.11E-01 | −0.114 | 2.93E-03 | 0.280 | 21.4 | 0.84 |
| rs11942832 | 4 | 24951385 | CCDC149 | intronic | T | C | 0.309 | −0.133 | 1.06E-06 | −0.097 | 4.89E-02 | −0.156 | 6.14E-03 | −0.147 | 2.71E-04 | 0.661 | 0 | 0.88 |
| rs118050709 | 4 | 39178700 | KLHL5/WDR19 | intergenic | C | A | 0.046 | −0.297 | 9.06E-07 | −0.355 | 3.15E-03 | −0.267 | 2.72E-02 | −0.283 | 9.91E-04 | 0.851 | 0 | 0.92 |
| rs1289351462 | 5 | 60624566 | LINC02057/ZSWIM6 | intergenic | A | ACAATGGCTTAGG | 0.007 | −0.953 | 2.24E-06 | −1.419 | 4.42E-04 | −0.567 | 2.04E-01 | −0.885 | 1.16E-03 | 0.343 | 6.7 | 0.50 |
| rs2243036 | 5 | 76122388 | F2RL1 | intronic | G | A | 0.251 | 0.132 | 6.54E-06 | 0.087 | 8.12E-02 | 0.132 | 3.33E-02 | 0.168 | 1.61E-04 | 0.475 | 0 | 0.86 |
| rs61093400 | 5 | 108415405 | FER | intronic | A | G | 0.027 | 0.373 | 3.98E-06 | 0.195 | 3.93E-01 | 0.413 | 7.77E-04 | 0.385 | 1.58E-03 | 0.695 | 0 | 0.46 |
| rs10749438 | 10 | 123810832 | TACC2 | intronic | G | A | 0.706 | −0.148 | 4.24E-08 | −0.143 | 3.99E-03 | −0.140 | 9.94E-03 | −0.156 | 9.81E-05 | 0.966 | 0 | 0.93 |
| rs6483495 | 11 | 96036188 | MAML2 | intronic | G | A | 0.310 | −0.140 | 5.40E-08 | −0.113 | 1.88E-02 | −0.123 | 2.21E-02 | −0.164 | 9.45E-06 | 0.665 | 0 | 0.99 |
| rs182016826 | 12 | 116219902 | TBX3/MED13L | intergenic | T | C | 0.017 | −0.453 | 4.99E-06 | −0.719 | 2.71E-05 | −0.278 | 1.58E-01 | −0.343 | 2.61E-02 | 0.157 | 46 | 0.88 |
| rs117436582 | 13 | 37367772 | SERTM1/RFXAP | intergenic | A | G | 0.050 | −0.266 | 7.27E-06 | −0.216 | 2.26E-02 | −0.300 | 2.82E-02 | −0.299 | 1.14E-03 | 0.788 | 0 | 0.84 |
| rs76373752 | 14 | 34645522 | EGLN3/SPTSSA | intergenic | T | C | 0.042 | 0.305 | 5.38E-06 | 0.366 | 2.09E-03 | 0.223 | 8.13E-02 | 0.314 | 2.90E-03 | 0.711 | 0 | 0.77 |
| rs3743680 | 16 | 69153608 | CHTF8 | 3'UTR | C | T | 0.027 | −0.393 | 7.40E-06 | −0.509 | 3.76E-04 | −0.509 | 7.69E-03 | −0.228 | 9.33E-02 | 0.289 | 19.4 | 0.68 |
| rs148814682 | 18 | 69057390 | GTSCR1/LINC01541 | intergenic | C | T | 0.036 | 0.304 | 3.51E-06 | 0.289 | 1.52E-02 | 0.448 | 4.22E-03 | 0.265 | 3.58E-03 | 0.591 | 0 | 0.98 |
| rs1718047 | X | 32179025 | DMD | intronic | C | A | 0.851 | 0.136 | 2.92E-06 | 0.209 | 5.28E-05 | 0.066 | 2.86E-01 | 0.120 | 5.24E-02 | 0.396 | 3.2 | 0.99 |

Chr chromosome, Ref reference allele, Alt alternative allele, META meta-analysis, Freq allele frequency of an alternative allele, Beta beta of an alternative allele, P_het P-value for Cochran's Q-test of heterogeneity, INFO imputation quality score.

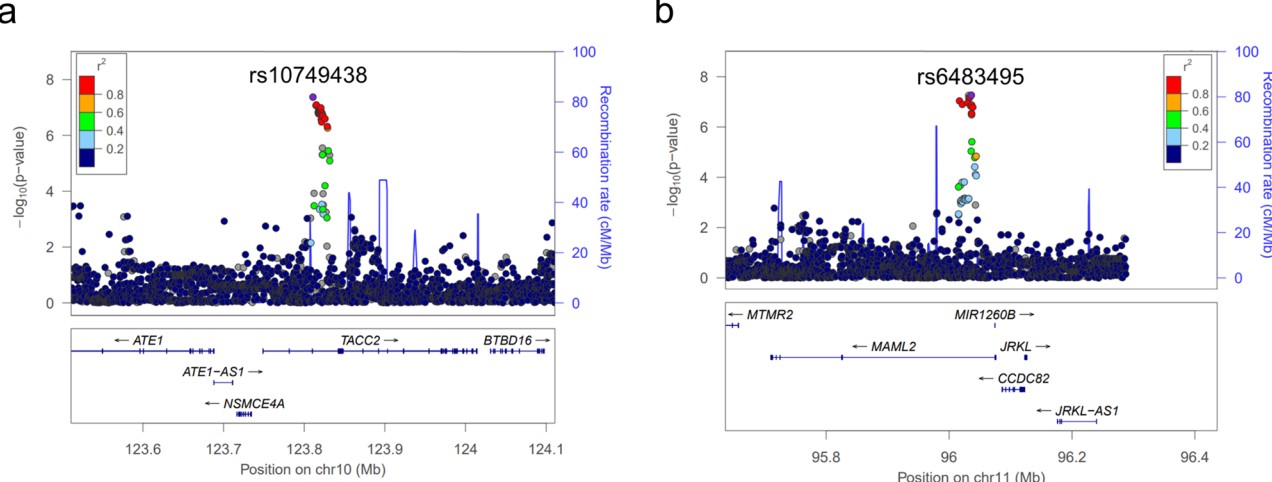

**Fig. 2 | Regional plot. a** The significant locus with the lead variant (rs10749438) and (**b**) the suggestive locus with the lead variant (rs6483495) associated with knee extension strength.

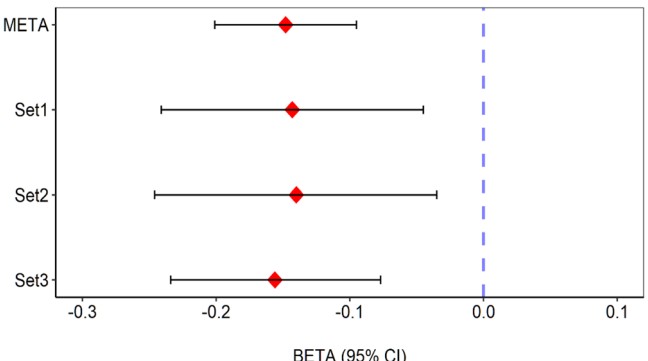

**Fig. 3 | Forest plot of the lead variant (rs10749438).** Consistent associations were observed across the three datasets. META, the effect size of meta-analysis. Error bar, 95% confidence interval. Beta are shown in Table 2.

was involved in tumorigenesis of a variety of cancers[30,44–46]. These findings suggest TACC2 may function in muscle via modulating cell division. Further studies would be necessary to clarify the role of TACC2 in muscle strength and sarcopenia.

We also identified 17 suggestive loci including candidate causal variants. While these variants are good candidates for further replication analyses, we should be cautious of variants with low minor allele frequencies due to possible inaccurate imputation compared with common variants. That is another limitation of the study and future studies with larger dataset will be needed to confirm the associations. We investigated if those suggestive variants are associated with sarcopenia-related traits such as lean body mass, frailty, walking pace fatigue, testosterone and IGF1 in the UK Biobank. We did not find very consistent patterns of associations (Supplementary Note), suggesting that ancestry matching for GWAS and further expansion of sample size for muscle strength is necessary.

In summary, we identified a novel locus associated with knee extension strength. This finding provides insights into the genetic architectures underlying muscle strength in lower limbs. It would be interesting to integrate the current results with studies of sarcopenia in the future.

## Methods
### Shimane CoHRE study
This cross-sectional study is a part of the cohort study conducted by the Center for Community-based Healthcare Research and Education in Shimane University (Shimane CoHRE Study)[19,20]. Shimane CoHRE Study is an

ongoing health examination for the community-dwelling people in Shimane prefecture, Japan. It complied with all relevant ethical regulations. The study protocol was approved by the Ethics Committee of Shimane University School of Medicine. Written informed consent was obtained from all participants. Based on different recruitment and genotyping terms, two data sets, Shimane 1st cohort (Set 1) and Shimane 2nd cohort (Set 2) were obtained. The participants of Set 1 and Set 2 were analyzed separately for those aged 60 years or older and those aged under 60 years. The characteristics of the participants were shown in Table 1 and Supplementary Table 5. The participants were all Japanese. We did not exclude any participants with knee osteoarthritis in the study.

### Bunkyo Health Study
Bunkyo Health Study is a prospective cohort study of over 10 years[21], which recruited older subjects aged 65–84 years living in Bunkyo-Ku, an urban area in Tokyo, Japan. Among the 68 communities in Bunkyo-Ku, we selected 13 communities based on probability proportionate to size sampling. We obtained the name and address of all residents aged 65–84 years in the selected communities from residential registries. The exclusion criteria were to have a pacemaker or defibrillator placement and diabetes mellitus requiring insulin therapy. All participants provided written informed consent. The details of the characteristics were shown in Table 1. The participants were all Japanese. We did not exclude any participants with knee osteoarthritis in the study.

### Phenotype
In Shimane CoHRE Study, knee extension strength was measured by using the Quadriceps Training Machine (QTM) (QTM-05F, Alcare, Tokyo, Japan). The device has a knee holding part corresponding to the knee joint with approximately 30° flexion. Participants were asked to put some muscle as hard as possible for three seconds, and the maximum value that was reached during that time period was recorded. Both legs were measured in turn. We calculated relative knee extension strength as an average of measurements of the right and left legs divided by weight. The average of measurements was regressed and residualized by age and sex, and the residuals were inverse-rank normalized and used as quantitative phenotypes.

In Bunkyo Health Study (Set 3), knee extension muscle strength was measured by using the BIODEX system 4 (Biodex Medical Systems, Upton, New York, USA), which measures isokinetic knee muscle strength. To measure a value close to the maximum extension muscle force, we adopted the maximum torque at an angular velocity of 60°. As in Set 1 and Set 2, we

**Table 3 | Age and sex-stratified analyses of rs10749438**

| rsID | Chr | Position | Gene | Location | Ref | Alt | Subjects | N | Freq | Beta | SE | P-value |
|------|-----|----------|------|----------|-----|-----|----------|---|------|------|-----|---------|
| rs10749438 | 10 | 123810832 | *TACC2* | intronic | G | A | ≥60 y.o. | 3452 | 0.706 | −0.148 | 0.027 | 4.24E-08 |
| | | | | | | | male | 1224 | 0.707 | −0.131 | 0.045 | 3.67E-03 |
| | | | | | | | female | 2017 | 0.705 | −0.164 | 0.035 | 3.14E-06 |
| | | | | | | | <60 y.o. | 173 | 0.687 | −0.200 | 0.123 | 1.05E-01 |
| | | | | | | | ≥75 y.o. | 1296 | 0.700 | −0.170 | 0.043 | 7.34E-05 |

male: male participants aged 60 year or older, female: female participants aged 60 years or older.

*Chr* chromosome, *Ref* reference allele, *Alt* alternative allele, *N* number of samples, *Freq* allele frequency for an alternative allele, *Beta* beta of an alternative allele, *SE* standard error, *y.o.* years old.

calculated relative knee extension strength as an average of measurements of the right and left legs divided by weight. The average of measurements was regressed and residualized by age and sex, and the residuals were inverse-rank normalized and used as quantitative phenotypes.

### Genotyping and quality control
We genotyped samples of Set 1 with the Illumina HumanOmniExpressExome BeadChip and those of Set 2 and Set 3 with the Illumina Asian Screening Array.

For quality control of samples, we excluded those with: (1) sex inconsistency between genotype and clinical data, (2) genetically identical to others (PI_HAT > 0.9, PI_HAT was based on identity by decent (IBD), i.e., P(IBD = 2) + 0.5*(IBD = 1)), (3) sample call rate < 0.98, and (4) outliers from East Asian clusters identified by PC analysis using genotypes in the HAPMAP project. For quality control of genotypes, we excluded variants meeting any of the following criteria: (1) call rate < 0. 99, (2) Hardy–Weinberg equilibrium $p < 1.0 \times 10^{-6}$, (3) the allele frequency show difference between the reference > = 6% compared with the reference panel.

### Imputation
We utilized the 1000 Genomes Project Phase 3 [1KGP3v5; (May 2013, $n = 2504$)] and 3256 in-house Japanese whole-genome sequence data obtained from the Biobank Japan[47] (JEWEL_3K) for imputation to achieve better imputation accuracy for the Japanese population as previously described[48]. In brief, samples were sequenced at high depth (15x, 30x) on various platforms. The whole-genome sequencing data was processed, following the standardized best practice method in Genome Analysis Toolkit (GATK). In addition to the process of the best practice, we put additional filters of approximate read depth and genotype quality before variant quality score recalibration (VQSR). The variants at multi-allelic sites were removed from the combined reference panel by vcftools (version 0.1.14). We estimated the haplotypes by SHAPE IT (version 2.778) and combined the data of the 1KG phase 3v5 and the BBJ by using IMPUTE2[49,50]. Quality control was performed with bcftools (version 1.3.1) and vcftools (version 0.1.14). Variants at multi-allelic sites, monomorphic sites and singletons were excluded. We performed pre-phasing using EAGLE2.4.1 (https://alkesgroup.broadinstitute.org/Eagle/) to determine the haplotypes. We imputed the genotype dosages with minimac4 (v1.0.0)[51]. After imputation, we excluded variants with an imputation quality of Rsq < 0.3 and minor allele frequency <0.005. Imputation quality of Rsq is the estimated value of the squared correlation between imputed genotypes and true, unobserved genotypes.

### GWAS and meta-analysis
We conducted GWAS using the fastGWA[23] linear mixed model package and used the top three PCs as covariates. We excluded variants with minor allele frequencies <0.005. We conducted GWASes separately for three sets and performed an inverse variance fixed-effects meta-analysis by using METAL[24]. For the X chromosome, we performed GWAS in males and females separately and meta-analyzed using METAL[24]. METAL also calculates $I^2$ which describes the percentage of variation across studies that is due to heterogeneity rather than chance. We annotated the variants which exceeded the significant threshold ($P < 5 \times 10^{-8}$) and the suggestive threshold ($P < 1 \times 10^{-5}$) in the GWAS by using ANNOVAR[52], HaploReg[26] and ENCODE database[25].

### LDSC
We estimated the heritability of the knee extension strength GWAS result using LDSC (version 1.0.0). We excluded variants in the human leukocyte antigen region (chromosome 6: 26–34 Mb). We further calculated heritability z-scores and standard errors (SEs) to assess the reliability of heritability estimation. The heritability is based on the variants in additive model[53].

### Age- and sex-stratified analyses
To investigate the effect of age in the association of significant and suggestive variants in the participants aged 60 years or older, we conducted age-stratified analyses, including participants aged under 60 years and those aged 75 years or older. Since the linear mixed model did not converge due to the small sample size in these analyses, we performed a linear regression by using PLINK 2.0 after excluding related individuals (PI_HAT > 0.25, Supplementary Fig. 8). We also conducted sex-stratified analyses by using PLINK 2.0 in the same manner. We showed the results of statistical power analyses in Supplementary Fig. 10.

### Bayesian statistical fine-mapping analysis
We performed statistical fine-mapping analysis using FINEMAP software (version 1.3.1)[54] to prioritize causal variants in susceptible loci. The FINEMAP computes a posterior probability of causality for each variant. We ranked candidate putative causal variants in a descending order of their posterior probabilities and created a 95% credible set of causal variants by adding the posterior probabilities of the ordered variants until their cumulative posterior probabilities reached 0.95. We used the default priors and parameters in FINEMAP.

### Evaluation of the variants identified in a GWAS of handgrip strength
We investigated if 140 variants identified in the GWAS of handgrip strength[15] had the same direction of effect in our GWAS. Since our dataset includes only 108 variants out of those variants, we used 12 high LD variants (r2 > 0.8) with other variants. We calculated LD based on 1KG European ancestry data. We conducted a binomial test; 120 variants were tested and 0.5 was the expected proportion of variants with the same direction of effect.

### Statistics and reproducibility
We did not perform any statistical method to predetermine sample size because we used all available samples we have to maximize statistical power. GWAS were performed by using fastGWA[23] linear mixed model package and used the top three PCs as covariates. A meta-analysis was performed by using METAL[24]. Significant threshold of the GWAS meta-analysis is $p \le 5 \times 10^{-8}$ accounting for multiple testing. For the evaluation of the variants identified in a GWAS of handgrip strength, we conducted binomial test by using R (version 4.0.2).

### Reporting summary
Further information on research design is available in the Nature Portfolio Reporting Summary linked to this article.

**Table 4 | The meta-analysis for 3625 participants aged 60 years or older and those aged under 60 years**

| rsID | Chr | Position | Gene | Location | Ref | Alt | Freq | Beta | P-value | $P_{het}$ | $I^2$ |
|---|---|---|---|---|---|---|---|---|---|---|---|
| rs141279361 | 1 | 202923428 | ADIPOR1 | 5′UTR | C | T | 0.026 | −0.336 | 1.16E-05 | 0.327 | 0 |
| rs77607073 | 2 | 49531177 | FSHR/NRXN1 | intergenic | A | G | 0.115 | 0.184 | 8.60E-07 | 0.398 | 0 |
| rs147964289 | 3 | 3825426 | CRBN/LRRN1 | intergenic | A | C | 0.012 | −0.509 | 1.59E-05 | 0.164 | 48.3 |
| rs14161911 | 3 | 105393996 | CBLB | intronic | T | C | 0.008 | 0.622 | 1.48E-05 | 0.181 | 44 |
| rs138068168 | 3 | 158936130 | IQCJ,IQCJ-SCHIP1 | intronic | T | TTG | 0.475 | −0.115 | 5.01E-06 | 0.962 | 0 |
| rs11942832 | 4 | 24951385 | CCDC149 | intronic | T | C | 0.309 | −0.132 | 8.10E-07 | 0.746 | 0 |
| rs118050709 | 4 | 39178700 | KLHL5/WDR19 | intergenic | C | A | 0.048 | −0.286 | 9.38E-07 | 0.485 | 0 |
| rs1289351462 | 5 | 60624566 | LINC02057/ZSWIM6 | intergenic | A | ACAATGGCTTAGG | 0.007 | −0.952 | 2.31E-06 | 0.752 | 0 |
| rs2243036 | 5 | 76122388 | F2RL1 | intronic | G | A | 0.253 | 0.137 | 1.69E-06 | 0.496 | 0 |
| rs61093400 | 5 | 108415405 | FER | intronic | A | G | 0.031 | 0.345 | 1.11E-05 | 0.151 | 51.4 |
| rs10749438 | 10 | 123810832 | TACC2 | intronic | G | A | 0.705 | −0.151 | 1.16E-08 | 0.681 | 0 |
| rs6483495 | 11 | 96036188 | MAML2 | intronic | G | A | 0.310 | −0.131 | 2.16E-07 | 0.087 | 66 |
| rs182016826 | 12 | 116219902 | TBX3/MED13L | intergenic | T | C | 0.018 | −0.432 | 7.07E-06 | 0.396 | 0 |
| rs117436582 | 13 | 37367772 | SERTM1/RFXAP | intergenic | A | G | 0.052 | −0.260 | 7.88E-06 | 0.602 | 0 |
| rs76373752 | 14 | 34645522 | EGLN3/SPTSSA | intergenic | T | C | 0.043 | 0.299 | 4.56E-06 | 0.702 | 0 |
| rs3743680 | 16 | 69153608 | CHTF8 | 3′UTR | C | T | 0.027 | −0.373 | 1.42E-05 | 0.249 | 24.9 |
| rs148814682 | 18 | 69057390 | GTSCR1/LINC01541 | intergenic | C | T | 0.036 | 0.281 | 1.23E-05 | 0.076 | 68.3 |

*Chr* chromosome, *Ref* reference allele, *Alt* alternative allele, *Freq* allele frequency of an alternative allele, *Beta* beta of an alternative allele, $P_{het}$ P-value for Cochran's Q-test of heterogeneity.

## Data availability

The GWAS summary statistics generated in this study is available in the JENGER database and GWAS catalog (http://ftp.ebi.ac.uk/pub/databases/gwas/summary_statistics/GCST90319001-GCST90320000/GCST90319502). The remaining data are available with in the article, Supplementary Information and Source Data file. The source data behind the graphs in the paper can be found in Supplementary Data 1.

## Code availability

The code of statistical analyses is available on GitHub (URL: https://github.com/Shuji2022/Code) and is also archived in Zenodo (URL: https://doi.org/10.5281/zenodo.10675274)[55].

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

## Acknowledgements

We thank all the participants of the Shimane CoHRE Study and the Bunkyo Health Study. Furthermore, we would like to express our gratitude to members of the studies for their skillful assistance and data collection. This work is supported by JSPS KAKENHI (18H03184, 19H03996, 20H00462, 22H03207) and the Strategic Research Foundation at Private Universities

(S1411006) from the Ministry of Education, Culture, Sports, Science and Technology of Japan, the Mizuno Sports Promotion Foundation, the Mitsui Life Social Welfare Foundation, AMED (JP21ek0109555, JP21tm0424220 and JP21ck0106642, JP23ek0410114, JP23tm0424225), and Takeda Hosho Grants for Research in Medicine.

## Author contributions

S.Ito, C.T., and S.Ikegawa designed the study. S.Ito analyzed the data with the help of Y.I., X.L., N.O., H.S., Y.K., K.H., K.T., S.Ikegawa, and C.T.; S.Ito wrote the manuscript. Y.M. and K.O. performed the genotyping for the GWAS. S.Ito, H.T., S.Kakehi, Y.S., H.Kaga, N.K., S.Kuwata, T.W., M.K., S.Y., T.A., M.T., M.Isomura, T.N., H.K., M.Ishijima, R.K., H.W., Y.T. and Y.U. collected and managed DNA samples and clinical data. All authors critically reviewed and revised the manuscript draft and approved the final version for submission.

## Competing interests

The authors declare no competing interests.

## Additional information

[1]Laboratory for Bone and Joint Diseases, RIKEN Center for Integrative Medical Sciences, Tokyo 108-8639, Japan. [2]Laboratory for Statistical and Translational Genetics, RIKEN Center for Integrative Medical Sciences, Yokohama 230-0045, Japan. [3]Department of Orthopedic Surgery, Shimane University Faculty of Medicine, Izumo 693-8501, Japan. [4]Department of Metabolism & Endocrinology, Juntendo University Graduate School of Medicine, Tokyo 113-8421, Japan. [5]Sportology Center, Juntendo University Graduate School of Medicine, Tokyo 113-8421, Japan. [6]Graduate School of Health and Sports Science, Juntendo University, Inzai 270-1695, Japan. [7]Department of Orthopedic Surgery, Matsue Red Cross Hospital, Matsue 690-8506, Japan. [8]The Center for Community-based Healthcare Research and Education (CoHRE), Shimane University, Izumo 693-8501, Japan. [9]Department of Orthopaedic Surgery, School of Medicine, Keio University, Tokyo 160-8582, Japan. [10]Department of Orthopaedic Surgery, Graduate School of Medical Sciences, Kyushu University, Fukuoka 812-8582, Japan. [11]Department of Orthopedic Surgery, Hokkaido University Graduate School of Medicine, Sapporo 060-8638, Japan. [12]Laboratory for Pharmacogenomics, RIKEN Center for Integrative Medical Sciences, Yokohama 230-0045, Japan. [13]Laboratory for Genotyping Development, RIKEN Center for Integrative Medical Sciences, Yokohama 230-0045, Japan. [14]Medical Genome Center, Research Institute, National Center for Geriatrics and Gerontology, Obu 474-8511, Japan. [15]Faculty of Human Sciences, Shimane University, Matsue 690-8504, Japan. [16]Department of Functional Pathology, Shimane University School of Medicine, Izumo 693-8501, Japan. [17]Department of Medicine for Orthopaedics and Motor Organ, Juntendo University Graduate School of Medicine, Tokyo 113-8421, Japan. [18]Clinical Research Center, Shizuoka General Hospital, Shizuoka 420-8527, Japan. [19]The Department of Applied Genetics, The School of Pharmaceutical Sciences, University of Shizuoka, Shizuoka 422-8526, Japan. ✉e-mail: chikashi.terao@riken.jp

