## [Peer Review File · Communications Biology]

Reviewers' comments:

Reviewer #1 (Remarks to the Author):

This is a well-written manuscript aimed to identify genetic variants associated with lower limb muscle strength. The findings are novel and may be of interest to exercise scientists and geriatricians.

I have some comments.

Abstract. The lead variant (i.e. rs10749438) has to be mentioned in the abstract.

Introduction. Lines 92-93. Please indicate the exact number of variants from three studies. I think there should be at least 170 variants, as reported here: <https://pubmed.ncbi.nlm.nih.gov/36771461/>

Results. Lines 140-143. Please indicate which allele is a risk (A?) or protective (G?) allele.

Results. Using summary statistics data (for example, <https://genetics.opentargets.org/> or other databases), please check if TACC2 variant is nominally ($p < 0.05$) associated with other sarcopenia-related traits in the UK Biobank / FinnGen such as lean mass, frailty, falls, walking pace, fatigue, testosterone, IGF1 etc. If so, is direction of association the same? Please discuss these (positive or negative) findings in the Discussion. I also recommend to do this with 17 suggestive SNPs.

Results. Lines 179-185. I recommend to check all 170 variants (not 140) from three GWASes.

Results. Line 183. "Among the 71 variants, five showed an association..." Was the direction of association the same? Were these five variants sarcopenia-related (i.e. from the list of 78 SNPs: <https://pubmed.ncbi.nlm.nih.gov/36771461/>)?

Reviewer #2 (Remarks to the Author):

To further discern the genetic architecture of sarcopenia and muscle deterioration with age, the authors conducted a genome-wide analysis of lower limb muscle strength and identified several candidate loci. They used some metrics of knee extension strength in a relatively modest sample of 3,452 60+ years old Japanese. Significant association with TACC2 was discovered. This cytoskeleton-related protein is highly expressed in skeletal muscle and was reported as a causal gene in etiology of myotonic dystrophy 1. The authors thus suggest TACC2 as a potential therapeutic target for sarcopenia. No functional (experimental) validation of this assumption was attempted, while realize the gene candidacy requires a wet-lab / animal model validation.

This is a study by a group of experts in genetic epidemiology and biostatistics. The general idea of the manuscript is state-of-the-art, and has a potential value for a biological research in the muscle aging. Use of a relatively homogenous ethnic population for GWAS meta-analysis is a plus. This original study could

be a timely and interesting contribution. However, there are some concerns which to me make this work less impactful/unfinished in its current format.

The lead variant is rs10749438, an intronic variant in TACC2, is not an expression quantitative trait locus (eQTL) for TACC2. This is concerning. The authors can be referred to a paper by Claussnitzer 2015 (PMID: 26287746) where intronic variants in FTO actually regulated expression patterns in nearby IRX3 and IRX5, but not FTO itself.

Next, inverse variance fixed-effects meta-analysis was performed. However, given that there are differences among studies in the phenotype definitions, the random-effects model should fit the reality better. What were findings of that model?

There are several additional points, technical issues and language errors, which should be re-assessed by the authors in order to make this paper more original and impactful.

Introduction

I. 98: "GWASs of lower limb muscle strength would add better information..." is (a) repetitive from the previous sentences and (b) "better" than what?

Results

Pls. remind what the intercept of 1.00 means in the LDSC.

II. 161-162: the point re: "larger sample size necessary" belongs to the Discussion.

Methods:

No mention of the knee OA or joint replacement as exclusions is provided.

In Bunkyo Health Study (Set 3), the residual relative knee extension strength was not regressed on weight, correct?

Pls. justify why the threshold of difference between the patient's and reference panel's allele frequency $\geq 6\%$ was chosen.

II.300 and 305 both repeat the MAF-based exclusions.

Participants we excluded (I. 277) for being genetically identical to others ($PI_HAT > 0.9$) – then we see $PI_HAT > 0.25$ on I. 323.

Discussion:

The sentence "large part of genetic architecture may be different in muscle strength of upper and lower limbs" is problematic, - first, what is "large" in quantitative terms; second, this study couldn't solve that question by design.

"confounding effect of knee osteoarthritis" is mentioned only in Discussion, - how was this done (Mendelian Randomization? Adjustment?)

Minor points:

Unclear, what is an "enhanced association"; "direction of rs1718074" is a jargon. Abbrev. Rsq should be

expanded here; also, BBJ – either expand or remove it.

l. 215: pls. remove some extra letter. L. 275: should be “illuminA”.

Table 1:

Mean muscle strength/Body weight (s.d.) – last column (Set3) - units in Nm/kg, - is this correct?

Response to Reviewers

To Reviewer #1

This is a well-written manuscript aimed to identify genetic variants associated with lower limb muscle strength. The findings are novel and may be of interest to exercise scientists and geriatricians.

RESPONSE: We thank the reviewer for the positive comments.

I have some comments.

Abstract. The lead variant (i.e. rs10749438) has to be mentioned in the abstract.

RESPONSE: Thank you very much for the suggestion. We mentioned the lead variant as follows in the Abstract.

(Line 60)

We identified a significant signal, rs10749438 which is an intronic variant in *TACC2* (transforming acidic coiled-coil-containing 2) ($P = 4.2 \times 10^{-8}$).

Introduction. Lines 92-93. Please indicate the exact number of variants from three studies. I think there should be at least 170 variants, as reported here: <https://pubmed.ncbi.nlm.nih.gov/36771461/>

RESPONSE: We thank the reviewer for pointing this out. We revised our statement to more accurately as reported. We modified the sentence as follows in the Introduction.

(Line 90)

Previous GWASes have discovered 170 variants associated with muscle strength¹⁵⁻¹⁹.

Results. Lines 140-143. Please indicate which allele is a risk (A?) or protective (G?) allele.

RESPONSE: Thank you very much for the feedback. We have indicated the risk allele for muscle weakness is A. We revised the manuscript as follows.

(Line 139)

We identified a novel locus significantly associated with knee extension strength (Figure 1 and Table 2). The lead variant is rs10749438, an intronic variant in *TACC2* (transforming acidic coiled-coil-containing 2) located at 10q26 (Beta = -0.15, $P = 4.2 \times 10^{-8}$) (Figure 2a and Table

2). The risk allele for muscle weakness is allele A.

Results. Using summary statistics data (for example, <https://genetics.opentargets.org/> or other databases), please check if TACC2 variant is nominally ($p < 0.05$) associated with other sarcopenia-related traits in the UK Biobank / FinnGen such as lean mass, frailty, falls, walking pace, fatigue, testosterone, IGF1 etc. If so, is direction of association the same? Please discuss these (positive or negative) findings in the Discussion. I also recommend to do this with 17 suggestive SNPs.

RESPONSE: We thank the reviewer for the valuable comments. We investigated if TACC2 variant is associated with other sarcopenia-related phenotypes. However, we could not find any nominal significant associations between the variant and those traits (lean body mass, frailty, walking pace, fatigue, testosterone and IGF1). These suggest that we need GWAS results for these traits in East Asians. In addition, we checked if all suggestive variants are associated with other sarcopenia-related traits. The suggestive variant, rs13022242 is nominally associated with lean body mass in the same direction. The suggestive variant, rs138068168, is nominally associated with testosterone in the same direction. rs13314849, which is in linkage disequilibrium with rs138068168, is nominally associated with frailty in the same direction. The suggestive variant, rs2243036 is also nominally associated with testosterone in the same direction. The suggestive variant, rs118050709 is nominally associated with IGF1 in the same direction.

We added the following sentences in the Discussion.

(Line 211)

We also investigated if rs10749438 is associated with other sarcopenia-related traits such as lean body mass³⁶, frailty³⁷, walking pace³⁸, fatigue³⁹, testosterone⁴⁰ and IGF1⁴¹ in the UK Biobank, which did not show any nominal significant association.

(Line 239)

We investigated if those suggestive variants are associated with sarcopenia-related traits such as lean body mass, frailty, walking pace fatigue, testosterone, and IGF1 in the UK Biobank. We did not find very consistent patterns of associations (Supplementary Note), suggesting that matching populations for GWAS and further expansion of sample size for muscle strength are necessary.

(Supplementary note)

We investigated if suggestive variants identified in the current study are associated with sarcopenia-related traits such as lean body mass, frailty, walking pace fatigue, testosterone and IGF1 in the UK Biobank. A suggestive variant, rs13022242 is nominally associated with lean

body mass in the same direction. A suggestive variant, rs138068168, is nominally associated with testosterone in the same direction. rs13314849, which is in linkage disequilibrium with rs138068168, is nominally associated with frailty in the same direction. A suggestive variant, rs2243036 is also nominally associated with testosterone in the same direction. A suggestive variant, rs118050709 is nominally associated with IGF1 in the same direction.

Results. Lines 179-185. I recommend to check all 170 variants (not 140) from three GWASes. **RESPONSE:** We thank the reviewer for the valuable suggestion. We have checked all 170 variants from three GWASes. As a result, 132 variants were included in our GWAS and 18 proxy variants were selected for the test based on the linkage disequilibrium. We did not find the rest 20 variants in EAS. We found that 87 variants out of those 150 variants showed the same direction and a binomial test p-value was 0.03. Among the 87 variants, five variants showed an association of nominal statistical significance. We modified the manuscript as follows.

(Line 177)

We further tested whether the variants associated with handgrip strength¹⁵⁻¹⁷ showed associations in our dataset. Out of 170 variants associated with three GWASes of handgrip strength, 132 were included in our dataset, and 18 proxy variants were selected for the test based on the linkage disequilibrium (LD) of Europeans. Among the 150 variants, 87 showed the same direction of effect and a binomial test p-value was 0.03. Among the 87 variants, five showed an association of nominal statistical significance (expected number: 4.35). These findings indicate that muscle strengths of upper and lower limbs share common genetic architecture.

Results. Line 183. "Among the 71 variants, five showed an association..." Was the direction of association the same? Were these five variants sarcopenia-related (i.e. from the list of 78 SNPs: <https://pubmed.ncbi.nlm.nih.gov/36771461/>)?

RESPONSE: Thank you very much for the valuable comments. As we revised above, "Among the 150 variants, 87 showed the same direction of effect and a binomial test p-value was 0.03. Among the 87 variants, five showed an association of nominal statistical significance (expected number: 3.75)." The direction of those five variants was the same. Two out of the five variants were sarcopenia-related.

To Reviewer #2 (Remarks to the Author):

To further discern the genetic architecture of sarcopenia and muscle deterioration with age, the authors conducted a genome-wide analysis of lower limb muscle strength and identified several candidate loci. They used some metrics of knee extension strength in a relatively modest sample of 3,452 60+ years old Japanese. Significant association with TACC2 was discovered. This cytoskeleton-related protein is highly expressed in skeletal muscle and was reported as a causal gene in etiology of myotonic dystrophy 1. The authors thus suggest TACC2 as a potential therapeutic target for sarcopenia. No functional (experimental) validation of this assumption was attempted, while realize the gene candidacy requires a wet-lab / animal model validation.

This is a study by a group of experts in genetic epidemiology and biostatistics. The general idea of the manuscript is state-of-the-art, and has a potential value for a biological research in the muscle aging. Use of a relatively homogenous ethnic population for GWAS meta-analysis is a plus. This original study could be a timely and interesting contribution. However, there are some concerns which to me make this work less impactful/unfinished in its current format.

RESPONSE: We thank the reviewer for the valuable comments and suggestions.

The lead variant is rs10749438, an intronic variant in TACC2, is not an expression quantitative trait locus (eQTL) for TACC2. This is concerning. The authors can be referred to a paper by Claussnitzer 2015 (PMID: 26287746) where intronic variants in FTO actually regulated expression patterns in nearby IRX3 and IRX5, but not FTO itself.

RESPONSE: We thank the reviewer for the comment. We agree that intronic variants are not always QTL loci for the nearest genes but other distant genes. While we could not identify any other distant genes that rs10749438 showed evidence of regulation, the other distant genes, *ATE1*, *NSMCE4A* and *BTBD16* are candidates to be regulated by rs10749438. Additionally, there is a possibility that the variant's functional effect is more context-dependent. We added the sentence as follows in the Discussion.

(Line 222)

Another possibility is that the variant regulates other distant genes, such as *ATE1*, *NSMCE4A* and *BTBD16*. These genes are also candidate of causal genes for knee extension strength. Additionally, there is a possibility that the variant's functional effect on TACC2 or another gene is more context-dependent and existing eQTL studies may not have detected such effects yet.

Next, inverse variance fixed-effects meta-analysis was performed. However, given that there are differences among studies in the phenotype definitions, the random-effects model should fit the reality better. What were findings of that model?

RESPONSE: We thank the reviewer for suggesting the valuable advice. We conducted the random-effect meta-analysis, which showed the significant association of rs10749438 (Beta = -0.148, $P = 4.2 \times 10^{-8}$) as the fixed-effects meta-analysis showed.

(Line 203)

The association was the same as a meta-analysis with the random effect model (Beta = -0.148, $P = 4.2 \times 10^{-8}$).

There are several additional points, technical issues and language errors, which should be re-assessed by the authors in order to make this paper more original and impactful.

Introduction

I. 98: “GWASs of lower limb muscle strength would add better information...” is (a) repetitive from the previous sentences and (b) “better” than what?

RESPONSE: We thank the reviewer for pointing this out. We should have made this statement clearer to compare lower limb muscle strength with upper limb muscle strengths. We revised the sentence as below.

(Line 94)

However, there is only one GWAS of lower limb muscle strength, which did not identify any significant loci18 due to a lack of statistical power. GWASes of lower limb muscle strength with adequate sample size would add better information for understanding of the genetic architecture of sarcopenia rather than upper limb muscle strength including handgrip strength.

Results

Pls. remind what the intercept of 1.00 means in the LDSC.

RESPONSE: We thank the reviewer for the feedback. The LD Score regression intercept can be used to distinguish between inflation from true polygenic signal and bias. The intercept of 1.00 means that there is no confounding biases for the GWAS.

II. 161-162: the point re: “larger sample size necessary” belongs to the Discussion.

RESPONSE: Thank you very much for the suggestion. We removed the following sentence “indicating further increase in sample size is necessary to conclude this point.”

Methods:

No mention of the knee OA or joint replacement as exclusions is provided.

RESPONSE: We thank the reviewer for pointing this out. We did not exclude any participants with knee OA in the study. We have revised the manuscript in Methods as below.

(Line 261)

We did not exclude any participants with knee osteoarthritis in the study.

In Bunkyo Health Study (Set 3), the residual relative knee extension strength was not regressed on weight, correct?

RESPONSE: We apologize for making you confused. We calculated relative knee extension strength as an average of measurements of the right and left legs divided by weight. That is why the residual relative knee extension strength was not regressed on weight. We added the sentences as follows.

(Line 284)

As in Set 1 and Set 2, we calculated relative knee extension strength as an average of measurements of the right and left legs divided by weight. The average of measurements was regressed and residualized by age and sex, and the residuals were inverse-rank normalized and used as quantitative phenotypes.

Pls. justify why the threshold of difference between the patient's and reference panel's allele frequency $\geq 6\%$ was chosen.

RESPONSE: We thank the reviewer for pointing this out. This step was set because we would like to exclude possible genotyping errors supported by strong deviation from allele frequencies in other data sets. We excluded those variants empirically as reported in previous studies (PMID: 37461309, 37612283).

ll.300 and 305 both repeat the MAF-based exclusions.

RESPONSE: We apologize for the unnecessary repetition. We have removed the latter from the manuscript.

Participants we excluded (l. 277) for being genetically identical to others ($PI_HAT > 0.9$) –

then we see $PI_HAT > 0.25$ on l. 323.

RESPONSE: We thank the reviewer for the comments. We are afraid that we made you confused about the QC. For the fastGWA linear mixed model, we excluded participants being genetically identical to others. In the sub-analyses (Age- and sex-stratified analyses) that the linear mixed model did not converge due to the small sample size, we excluded the related individuals ($PI_HAT > 0.25$) for a linear regression.

Discussion:

The sentence “large part of genetic architecture may be different in muscle strength of upper and lower limbs” is problematic, - first, what is “large” in quantitative terms; second, this study couldn’t solve that question by design.

RESPONSE: We thank the reviewer for the suggestion. We agree with the reviewer that large is not suitable for the current study and the study could not solve that question. We revised the sentence using substantial part instead of large part as follows in the Discussion.

(Line 196)

Accordingly, our result suggests that a substantial part of genetic architecture may be different in muscle strength of upper and lower limbs.

“confounding effect of knee osteoarthritis” is mentioned only in Discussion, - how was this done (Mendelian Randomization? Adjustment?)

RESPONSE: Thank you very much for the comments. We conducted GWAS with the adjustment of knee OA, which showed the same association of TACC2 variant ($P = 3.7E-08$, $BETA = -0.147$). We added the revised manuscript as follows in the Discussion.

(Line 204)

We additionally investigated if there is a possible confounding effect of knee osteoarthritis on knee extension muscle strength by using knee osteoarthritis as an additional covariate, but we found no confounding effect of knee osteoarthritis ($Beta = -0.15$, $P = 3.7 \times 10^{-8}$).

Minor points:

Unclear, what is an “enhanced association”; “direction of rs1718074” is a jargon.

RESPONSE: We thank the reviewer for pointing out these unclear words..

We revised the sentence using increased association instead of enhanced association.

We modified the sentence as follows in the Result.

(Line 167)

The effect size and the effect direction of rs1718074 were consistent between males and females.

Abbrev. Rsq should be expanded here; also, BBJ – either expand or remove it.

RESPONSE: Rsq means R square. BBJ means BioBank Japan. As indicated, we expanded the abbreviations.

l. 215: pls. remove some extra letter. L. 275: should be “illumina”.

RESPONSE: We apologize for the error. We revised the manuscript.

Table 1:

Mean muscle strength/Body weight (s.d.) – last column (Set3) - units in Nm/kg, - is this correct?

RESPONSE: We thank the reviewer for pointing this out. This is correct. As we mentioned in the Methods, we used the BIODEX system 4 (Biodex Medical Systems, Upton, New York, USA), which measures isokinetic knee muscle strength.

Reviewers' comments:

Reviewer #1 (Remarks to the Author):

In the revised version of manuscript the Authors have added more detailed data and satisfactory answered to the queries.

Reviewer #2 (Remarks to the Author):

The authors have addressed my comments to major extent, and the manuscript is now more clearly and consistently written. However, there are some concerns which to me make this work far from being complete and thus rendering any contribution to the field. The main limitation here is an absence of either functional follow-up (of e.g. a prioritized transcript factor in bone experiments) or the identification of the proposed causal SNPs through the fine-mapping study, or refining a polygenic risk score for the phenotype of interest.

The study has some conceptual gaps and therefore interpretation that makes me uncomfortable, examples:

The authors state their findings indicate that "muscle strengths of upper and lower limbs share common genetic architecture", while in response to Rev. 2 (me) they state that "result suggests that a substantial part of genetic architecture may be different in muscle strength of upper and lower limbs". It's either or, cannot be both. Moreover, this study cannot respond this question by design, since the formal bivariate analysis or genetic correlations were not obtained.

Thus, a speculation that "GWASes of lower limb muscle strength with adequate sample size would add better information for understanding of the genetic architecture of sarcopenia rather than ... handgrip strength" is still unbased, not supported by analysis.

Re: frequency $\geq 6\%$ to exclude possible genotyping errors; this threshold is indeed "empirically reported in previous studies (PMID: 37461309, 37612283)" but still is a "rule of the thumb".

The manuscript is generally improved but the narration is still not fluent. A phrase "matching populations for GWAS" is still not self-explanatory.

Response to Reviewers

Reviewer #1 (Remarks to the Author):

In the revised version of manuscript the Authors have added more detailed data and satisfactory answered to the queries.

RESPNOSE: We thank the reviewer for the comments.

Reviewer #2 (Remarks to the Author):

The authors have addressed my comments to major extent, and the manuscript is now more clearly and consistently written. However, there are some concerns which to me make this work far from being complete and thus rendering any contribution to the field.

RESPONSE: Thank you very much for the valuable comments. We revised the manuscript as follows according to the comments.

The main limitation here is an absence of either functional follow-up (of e.g. a prioritized transcript factor in bone experiments) or the identification of the proposed causal SNPs through the fine-mapping study, or refining a polygenic risk score for the phenotype of interest.

RESPONSE: Thank you very much for the feedback. We agree that functional follow-up for the variant is favorable. This should be addressed in future studies. As the reviewer suggested, we investigated the fine-mapping study of the variant (rs10749438) and identified that rs10749438 showed the highest posterior probability. However, since the posterior probability (0.079 for rs10749438) did not restrict candidates of putative causal variants only to the lead variant, we cannot assert the lead variant as a causal variant. Refinement of polygenic risk scores is interesting, but we do not have enough sample size for the analyses. Since we will make public summary statistics of the current results, we believe that researchers in this field can use our summary statistics to compute PRS in their study subjects in the future.

We added a sentence to the revised manuscript as follows in the Result and the Discussion. We also added the method of fine-mapping in the Methods.

(Line 143)

A statistical fine-mapping analysis revealed rs10749438 with the highest posterior probability

(Supplementary Table 1).

(Line 214)

While the top posterior probability and overlapping with the enhancer region suggest rs10749438 as a promising candidate of a causal variant, functional follow-up is necessary to conclude this point. Regarding a responsible gene in this association, using cell cultures from human embryonic muscle, myotonic dystrophy 1-associated splicing alterations were significantly enriched in TACC2 which is one of cytoskeleton-related gene^{27,28}.

(Line 344)

Bayesian statistical fine-mapping analysis

We performed statistical fine-mapping analysis using FINEMAP software (version 1.3.1)⁵⁴ to prioritize causal variants in susceptible loci. The FINEMAP computes a posterior probability of causality for each variant. We ranked candidate putative causal variants in a descending order of their posterior probabilities and created a 95% credible set of causal variants by adding the posterior probabilities of the ordered variants until their cumulative posterior probabilities reached 0.95. We used the default priors and parameters in FINEMAP.

The study has some conceptual gaps and therefore interpretation that makes me uncomfortable, examples:

The authors state their findings indicate that “muscle strengths of upper and lower limbs share common genetic architecture”, while in response to Rev. 2 (me) they state that “result suggests that a substantial part of genetic architecture may be different in muscle strength of upper and lower limbs”. It’s either or, cannot be both. Moreover, this study cannot respond this question by design, since the formal bivariate analysis or genetic correlations were not obtained.

Thus, a speculation that “GWASes of lower limb muscle strength with adequate sample size would add better information for understanding of the genetic architecture of sarcopenia rather than ... handgrip strength” is still unbased, not supported by analysis.

RESPONSE: Thank you very much for pointing out this point. Please excuse us for making you confused. What we meant was that muscle strengths of upper and lower limbs share only a small part of genetic architecture (in spite of statistical significance) because we observed a significant but weakly shared (58%) direction of effects of the 150 variants between hand grip strength and muscle strength in lower limbs. We modified this part accordingly.

Meanwhile, a speculation that “GWASes of lower limb muscle strength with adequate sample size would add better information for understanding of the genetic architecture of sarcopenia rather than ... handgrip strength” was based on the recent studies reporting that knee extension

strength, a proxy of lower limb strength, is more strongly associated with performance-based sarcopenia compared to handgrip strength. Our ultimate goal is to predict sarcopenia. To do this, studies to analyze muscle strength in lower limbs would have advantages over hand grip because of the findings in the recent studies.

However, partly shared genetic architecture and comparison of correlation with sarcopenia between upper and lower limbs may make readers confused. Thus, we modified the sentence in the Introduction accordingly.

(Line 181)

These findings suggest that muscle strength of upper and lower limbs may share a small part of genetic architecture.

(Line 195)

Accordingly, our results suggest that only small fraction of genetic architecture is shared between muscle strength of upper and lower limbs.

(Line 94)

GWASes of lower limb muscle strength with adequate sample size would add good information for understanding of the genetic architecture of sarcopenia.

Re: frequency $\geq 6\%$ to exclude possible genotyping errors; this threshold is indeed “empirically reported in previous studies (PMID: 37461309, 37612283)” but still is a “rule of the thumb”.

RESPONSE: Thank you for the comments. This threshold could be changed in this study, but we did not find good reasons to modify it. Actually, if we applied 4% of threshold, we lost 227 variants. If we set 8% of threshold, we gained only 3 variants. Since 4% seems to be too strict considering sample size in this study (especially variants with minor allele frequencies near 0.5), we believe that 6%, the previously defined threshold, is reasonable in this study.

The manuscript is generally improved but the narration is still not fluent. A phrase “matching populations for GWAS” is still not self-explanatory.

RESPONSE: Thank you very much for pointing out the linguistic problems. We modified the manuscript as much as possible. We modified the phrase accordingly.

(Line 243)

We did not find very consistent patterns of associations (Supplementary Note), suggesting that ancestry matching for GWAS and further expansion of sample size for muscle strength is necessary.